# Quantitative analysis of electroporation-mediated intracellular delivery via bioorthogonal luminescent reaction
Shiqi Wang [1] ✉, Mariia V. Shcherbii[2], Sami-Pekka Hirvonen[3], Gudrun Silvennoinen[3], Mirkka Sarparanta[3] & Hélder A. Santos[1,4]

Efficient intracellular delivery is crucial for biotherapeutics, such as proteins, oligonucleotides, and CRISPR/Cas9 gene-editing systems, to achieve their efficacy. Despite the great efforts of developing new intracellular delivery carriers, the lack of straightforward methods for intracellular delivery quantification limits further development in this area. Herein, we designed a simple and versatile bioorthogonal luminescent reaction (BioLure assay) to analyze intracellular delivery. Our results suggest that BioLure can be used to estimate the amount of intracellularly delivered molecules after electroporation, and the estimation by BioLure is in good correlation with the results from complementary methods. Furthermore, we used BioLure assay to correlate the intracellularly-delivered RNase A amount with its tumoricidal activity. Overall, BioLure is a versatile tool for understanding the intracellular delivery process on live cells, and establishing the link between the cytosolic concentration of intracellularly-delivered biotherapeutics and their therapeutic efficacy.

Intracellular delivery (i.e., introducing membrane-impermeable drugs, proteins, nucleic acids, and nanomaterials into cells) is essential for a broad spectrum of medical research, ranging from fundamental causes of diseases to applied pharmaceutical sciences[1]. In fundamental studies related to certain diseases, critical gene and protein functions are routinely evaluated by cell transfection, i.e., intracellular delivery of plasmid DNA, siRNA, and mRNA. In applied pharmaceutical research, we have also witnessed a massive surge of interest in the intracellular delivery of biotherapeutics, leading to the first FDA-approved siRNA-based therapeutics (Onpattro®), and the successful launch of more than 1 billion mRNA-COVID vaccines from BioNTech/Pfizer and Moderna[2,3].

Despite the commercial success of mRNA vaccines, intracellular delivery remains a key challenge for the clinical applications of membrane-impermeable biotherapeutics. Among the fifty products approved by US Food and Drug Administration (FDA) in 2021, only two drugs (oligonucleotides) require intracellular delivery[4]. All the other forty-eight drugs either diffuse through cell membranes freely or exert their functions in the extracellular environment. Even for the mRNA-COVID vaccines already on the market, the intracellular delivery obstacle still limits the overall therapeutic outcomes[5], evidenced by the relatively high doses and common local or systemic side effects experienced by the recipients[6]. According to the literature, only 0.01% mRNA is estimated to reach cytoplasm and express proteins after cellular uptake[7]. Therefore, tremendous and continuous efforts are dedicated to improving the efficiency of intracellular delivery and maximizing the therapeutic efficacy of emerging biotherapeutics (including but not limited to oligonucleotides, proteins, and CRISPR-Cas gene-editing tools).

An essential problem in the intracellular delivery field is quantitative evaluation[8]. However, there is no gold standard for intracellular delivery efficiency evaluation up to date, which has become the bottleneck for the intracellular delivery field. Currently, the most common strategy is to evaluate the biological outcomes of the intracellularly delivered molecules, i.e., by measuring targeted gene expression levels, protein activities, and related cellular responses. Although these approaches reflect the final therapeutic outcomes of tested formulations, the results are not directly related to intracellular delivery efficiency, since the outcomes are associated with total cellular uptake and multiple downstream processes[9]. Another evaluation strategy is based on fluorescent labeling of the molecule-of-interest (MOI), and imaged by fluorescence microscopy. However, it is challenging to distinguish the MOI reaching the cytoplasm from those attaching to membranes or entrapped in endosomes[10]. In this case, an in-house super-resolution microscopy setup, quantitative imaging

[1]Drug Research Program, Division of Pharmaceutical Chemistry and Technology, Faculty of Pharmacy, University of Helsinki, FI-00014 Helsinki, Finland. [2]Institute of Biotechnology, University of Helsinki, FI-00014 Helsinki, Finland. [3]Department of Chemistry, Faculty of Science, University of Helsinki, FI-00014 Helsinki, Finland. [4]Department of Biomaterials and Biomedical Technology, The Personalized Medicine Research Institute (PRECISION), University Medical Center Groningen, University of Groningen, 9713 AV Groningen, The Netherlands. ✉e-mail: shiqi.wang@helsinki.fi

computational algorithms and extensive expertise are required to get a quantitative understanding[11–13]. Other approaches include split protein complementation, which enables quantification by simple fluorescence/luminescence readout upon intracellular delivery. In split protein complementation, one protein fragment from a reporter protein, such as GFP, ubiquitin, or luciferase, is conjugated to the MOI. The other complementary fragment is expressed in genetically engineered cells[9,14–16]. This method reports exclusively MOIs reaching the cytoplasm, but it requires relatively large tag conjugation (tens or even hundreds of amino acid residuals), which may significantly change the physiochemical properties of MOIs[17–20].

Bioorthogonal chemistry provides a unique solution to tackle the challenge of MOI labeling. It enables minimal labeling on MOIs, by conjugating a small handle, which could react with a tracer with complementary reactive groups in situ through bioorthogonal reactions without interfering with native biochemical processes[21]. Advantages of bioorthogonal reactions include biocompatibility, selectivity, and rapid kinetics, which enable instant reactions with high yields even at low concentrations[22]. This is crucial for intracellular delivery quantification since the process is dynamic and the intracellularly-delivered MOIs are very limited. Despite the advantages of bioorthogonal chemistry, few reports explore the potential of in situ labeling intracellularly delivered MOI with bioorthogonal reactions[23–25]. Ochocki et al. described using copper-catalyzed click reaction to monitor the intracellular delivered peptides[23]. In this case the disadvantage of copper-catalyzed reaction is nesesity for cell fixing and permeabilization to allow for the reactants to pass through. The compromised cell membrane permeability may induce payload leakage from endosomes to the cytoplasm, leading to false positive results. More recently, Peier et al. reported a pulse-chase strategy using copper-free strain-promoted alkyne-azide cycloadditions (SPAAC) reaction combined with the HaloTag technique to conjugate a fluorophore to intracellular proteins[24]. The MOIs linked with an azide handle react with a dibenzoazacyclooctyne (DBCO) complementary group anchored on HaloTag protein in the cytoplasm. Then another fluorophore with the same azide handle is introduced and reacts with the rest available DBCO, generating readout signals. Notably, the readout from this pulse-chase bioorthogonal strategy is inversely proportional to the translocated MOIs (turn-off assay), which unavoidably limits the sensitivity.

The condensation between cysteine and cyanobenzothiazole is a bioorthogonal reaction, which proceeds fast, selectively, and efficiently in living systems[26,27]. It exists naturally in firefly luciferin synthesis. The reaction constant is approximately $9.19\ \mathrm{M}^{-1}\ \mathrm{s}^{-1}$ under physiological conditions, several orders of magnitude higher than common bioorthogonal reactions (ca. $0.003\ \mathrm{M}^{-1}\ \mathrm{s}^{-1}$ for Staudinger ligation and $0.1\ \mathrm{M}^{-1}\ \mathrm{s}^{-1}$ for SPAAC reaction)[28]. The condensation reaction has been successfully used by protein labeling in E. coli[29] and mammalian cells[30]. Furthermore, the reaction between D-cysteine (the unnatural stereoisomer of L-cysteine) with cyanobenzothiazole generates D-luciferin, which is the substrate of Firefly luciferase. In the presence of adenosine triphosphate (ATP), magnesium, and oxygen, the subsequent enzymatic oxidation of D-luciferin generates quantitative bioluminescence, with a quantum yield as high as 41%[31]. Due to the biocompatibility, high selectivity, and ease of bioluminescence detection, the two-step cascade reaction (first condensation and then luciferase-catalyzed luciferin oxidation) has been used in noninvasive imaging of protease activity in vitro and in vivo[32–34]. More recently, this strategy has also been used for peptide uptake studies[35], showing that bioluminescence output could be used in cellular and tissue permeability detection.

Herein, we designed a bioluminescent assay (BioLure) to quantify intracellular delivery by cascade bioorthogonal reactions in live cells (Fig. 1). The MOI is labeled with an unnatural amino acid, D-cysteine (Dcys) via a disulfide bond, which is susceptible to reduction in the cytoplasm (where the redox potential of cysteine/cystine is approximately −70 to −160 mV[36]). The reduction triggers the instant release of Dcys upon successful intracellular delivery. The released Dcys in the cytoplasm reacts with externally added 6-amino-2-cyanobenzothiazole, (NCBT, free diffusion to

cytoplasm[33]). The reaction product D-aminoluciferin (D-amLu) is a substrate of luciferase, thus triggering bioluminescence readout in luciferase-expressing cells detectable by a plate reader readily available in most labs.

Although similar reactions have been used by others for intracellular peptide internalization studies[35], only the relative delivery efficiencies were revealed, rather than the absolute concentration of the delivered cargo. The relative intracellular delivery efficiency evaluation methods could provide a comparative perspective of different delivery vehicles, whereas absolute quantification offers essential insights into the therapeutic dose of MOIs. Considering each MOI has its own optimal concentration range to exert its functions without cytotoxicity, the actual amount of intracellularly delivered MOI is highly valuable for therapeutic applications. In addition, previous reports focused on relatively small MOIs (peptides and small molecules) in cellular uptake studies[32,34,35]. However, this tool has never been explored on large MOIs with complex structures (such as proteins), which offer distinct advantages in therapeutic applications but are difficult to deliver.

In this proof-of-concept study, first we investigated the sensitivity and selectivity of BioLure assay in cell-free buffers and luciferase-expressing cells. Specifically, we optimized the reaction condition for the assay and tested model MOIs (dextran, lysozyme, and β-Galactosidase). We chose electroporation method for intracellular delivery because it is relatively simple (with well-defined, ready-to-use protocols), versatile (applicable to almost all cell types and MOIs), efficient and reproducible[37]. Furthermore, we studied the actual intracellular delivery dosage required for satisfying therapeutic outcomes using a functional protein MOI (RNase A).

## Results and discussion

To investigate the sensitivity and selectivity of the BioLure assay, we performed a preliminary test in a cell-free reaction buffer with all the reactants. Specifically, we added D-cystine (DcySS, a disulfide oxidized dimer of Dcysteine), NCBT, luciferase, ATP, and $Mg^{2+}$ to Hank's Balanced Salt Solution (HBSS buffer) using similar conditions reported in literature[33] (Fig. 2a). As shown in Fig. 2b, only the sample with NCBT, DcySS, and a reducing reagent TCEP (tris(2-carboxyethyl)phosphine) had a significant luminescence signal (>100-fold increase compared with the background). All the other groups had similar signal at the background level. This indicates that luminescence generation requires the presence of both D-cysteine in its reduced form and NCBT. Furthermore, the luminescence of the enantiomer L-cystine (LcySS) is similar to the background control, indicating the excellent selectivity of the reaction.

Next, we tested BioLure assay and the corresponding luminescence output on a luciferase-expressing human melanoma cell line (A375-Fluc-eGFP, Fig. 2c). Since luciferase, ATP and $Mg^{2+}$ required for the bioorthogonal reactions are already in the cytoplasm in excess[38], as well as reducing agents like glutathione (GSH); it is expected that DcySS will be cleaved to generate Dcys and react with NCBT, which is known to diffuse through cell membranes[33,35]. Notably, the intracellular GSH concentration is an important factor that could potentially affect the effectiveness of the system. According to literature reports, intracellular GSH concentration is within the millimolar range[39,40], which is far more excessive than the D-cysteine concentration used in this system (in the micromolar range). Thus, we assume that the GSH concentration is not the limiting factor of the reaction.

To verify the intracellular GSH concentration of A375 cells used in this study, we measured it using a commercial kit (GSH/GSSG-Glo™ assay, Promega). We evaluated both healthy cells and cells right after electroporation. The electroporated cells represent an extreme scenario when cells are exposed to transient membrane damage and subjected to GSH depletion[41]. The results suggested the GSH concentration within healthy A375 cells is 4.97 mM, which is similar to literature reports of cancer cell intracellular GSH concentrations measured by various other methods (e.g., 1.4 mM in HepG2 cells measured by HPLC[42]; 3.9-5.4 mM in Hela cells measured by fluorescent probes[43,44]; 6.1 mM in Hela cells measured by a single-cell nanopore sensor[45]. After electroporation, the GSH concentration decreased to 1.37 mM. It is still in micromolar range and much higher than the D-cysteine concentration. These results suggest that the

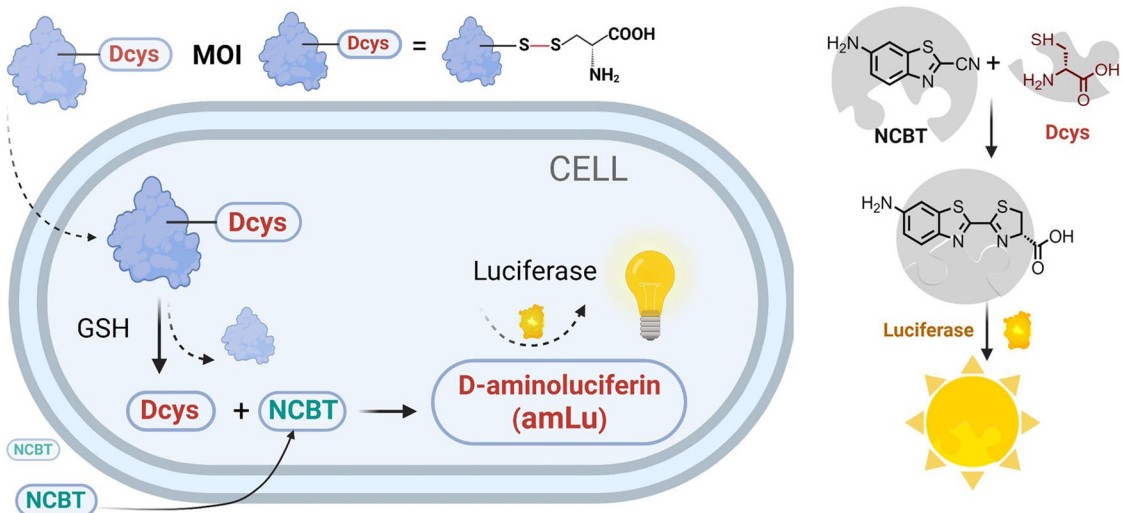

**Fig. 1 | The schematic showing of Biolur assay for intracellular delivery quantification and the key reactants involved.** MOI molecule of interest, GSH glutathione, Dcys D-cysteine, NCBT 6-amino-2-cyanobenzothiazole. Created with BioRender.com.

intracellular GSH concentration is significantly higher than D-cysteine to be analyzed by the BioLure assay.

To deliver DcySS into the cytoplasm in large amounts, cells were electroporated using Lonza nucleofector technology with a cell-specific pre-optimized protocol according to the manufacture's instruction. After electroporation, excessive NCBT was added, and the real-time luminescence was recorded. The electroporation procedure and the subsequent addition of NCBT did not significantly affect cell viability, as proved by 80% cell viability characterized by the intracellular ATP amount (Fig. S1). The luminescence output in DcySS sample was immediate after NCBT addition (Fig. S2), peaked after 5 min, and decreased slowly afterward. In contrast, the cells electroporated with LcySS only had background luminescence similar to the negative control. After integration of the real-time luminescence signal (from 0 to 30 min, Fig. 2d), we identified a significant signal output in the sample with both DcySS and NCBT. Unlike the results in reaction buffers (Fig. 2b), the addition of NCBT to A375-Fluc-eGFP cells without DcySS did induce background luminescence signals. This is probably due to a small amount of endogenous Dcys in the cytoplasm since it is a byproduct of cysteine metabolism[46]. Another possibility is the intracellular enantiomerization of L-aminoluciferin to D-aminoluciferin by a series of enzymes and coenzymes[47].

Next, we systematically studied how the bioorthogonal reaction conditions affect the luminescence output. Specifically, we investigated three key parameters: the number of cells (representing the available luciferase), NCBT and DcySS concentration. In each experiment, we fixed two parameters and varied the third one. As shown in Fig. 2e, the overall luminescence signal increased with the number of cells in the tested sample. The detection limit is *ca.* 200 cells, where the signal from DcySS containing sample is greater than three standard deviations above the background signal from Ctrl (with NCBT but without DcySS). Of note, the background of the Ctrl sample also increased with the number of cells in the sample, suggesting the endogenous Dcys or D-amLu after NCBT addition is cell-dependent. Figure 2f demonstrates the dependence of the luminescence signal on NCBT concentration. Since NCBT is the major reactant in the bioorthogonal reaction, increasing NCBT centration from 0.5 to 50 μM led to significantly higher luminescence output. Further increase in NCBT concentration to 100 μM led to a marginal increase in luminescence, and thus, 50 μM NCBT was selected for further evaluation. Regarding the available Dcys, Fig. 2g suggested a linear correlation between the average luminescence per cell and the Dcys concentration after removing the background signal from the negative control. This correlation makes it possible to use the

luminescence output to quantify the amount of Dcys in the cells after intracellular delivery.

After verifying BioLure's performance in the cell-free reaction buffer and on live luciferase-expressing cell lines, we decided to test BioLure using a model macromolecular MOI, dextran. Dextran (10 kDa) is a natural polysaccharide impermeable to cell membranes[48]. Due to its biocompatibility and simple structure, dextran has been widely used as a model payload for intracellular delivery studies[49,50]. In this study, we first labeled the Dcys tag on dextran using a succinimidyl 3-(2-pyridyldithio)propionate (SPDP) linker and then conjugate with Dcys (Fig. 3a). A control polymer Dex-Lcys (representing dextran labeled with L-form cysteine via the same conjugation strategy) was also synthesized using the same reaction strategy. Additionally, a non-cleavable dextran-Dcys conjugate polymer (Dex-NC) was synthesized using maleimide-thiol chemistry (Fig. 3a). The control polymers (Dex-Lcys and Dex-NC) were designed to test the reaction specificity.

The structures of all dextran conjugates were characterized by NMR spectroscopy (Figs. S3 and S4). In the NMR spectra, we compared the peaks from the starting polymer (amine-modified dextran), the intermediate (SPDP-conjugated dextran, and maleimide-conjugated dextran), as well as the final products. Representative peaks of SPDP and maleimide were identified after the first-step reaction and disappeared after the second-step reaction. The dextran conjugates were also characterized by elemental analysis (Table S1). The presence of sulfur in Dex-Dcys, Dex-Lcys, and Dex-NC indicates that cysteine was incorporated into the polymer structure after conjugation. On average, there are 3.1 Dcys residuals per dextran molecule, based on the release of pyridine 2-thione which has UV-absorbance at 343 nm. Then, all dextran conjugates were also labeled with Alexa647 fluorophore, making them detectable by fluorescence microscopy and flow cytometry. The Alexa647 labeling was confirmed by size exclusion chromatography (Fig. S5).

The successful intracellular delivery of dextran MOIs after electroporation was confirmed by confocal microscopy (Fig. 3b) and flow cytometry (Fig. S6). The red fluorescence from Dex-Dcys, Dex-Lcys, and Dex-NC colocalized well with eGFP, which is expressed in the cytoplasm of A375-Fluc-eGFP (Fig. 3b). The results from flow cytometry (Fig. S6) also confirmed that all the dextran MOIs have significantly higher mean fluorescence intensity (MFI) after electroporation, compared with non-electroporated controls. The final MFI of Dex-Dcys, Dex-Lcys, and Dex-NC samples are similar, suggesting a similar amount of dextran was delivered in these samples after electroporation. However, only Dex-Dcys generated detectable luminescence after electroporation and NCBT addition (Fig. 3c), while the other two control polymers have similar background luminescence

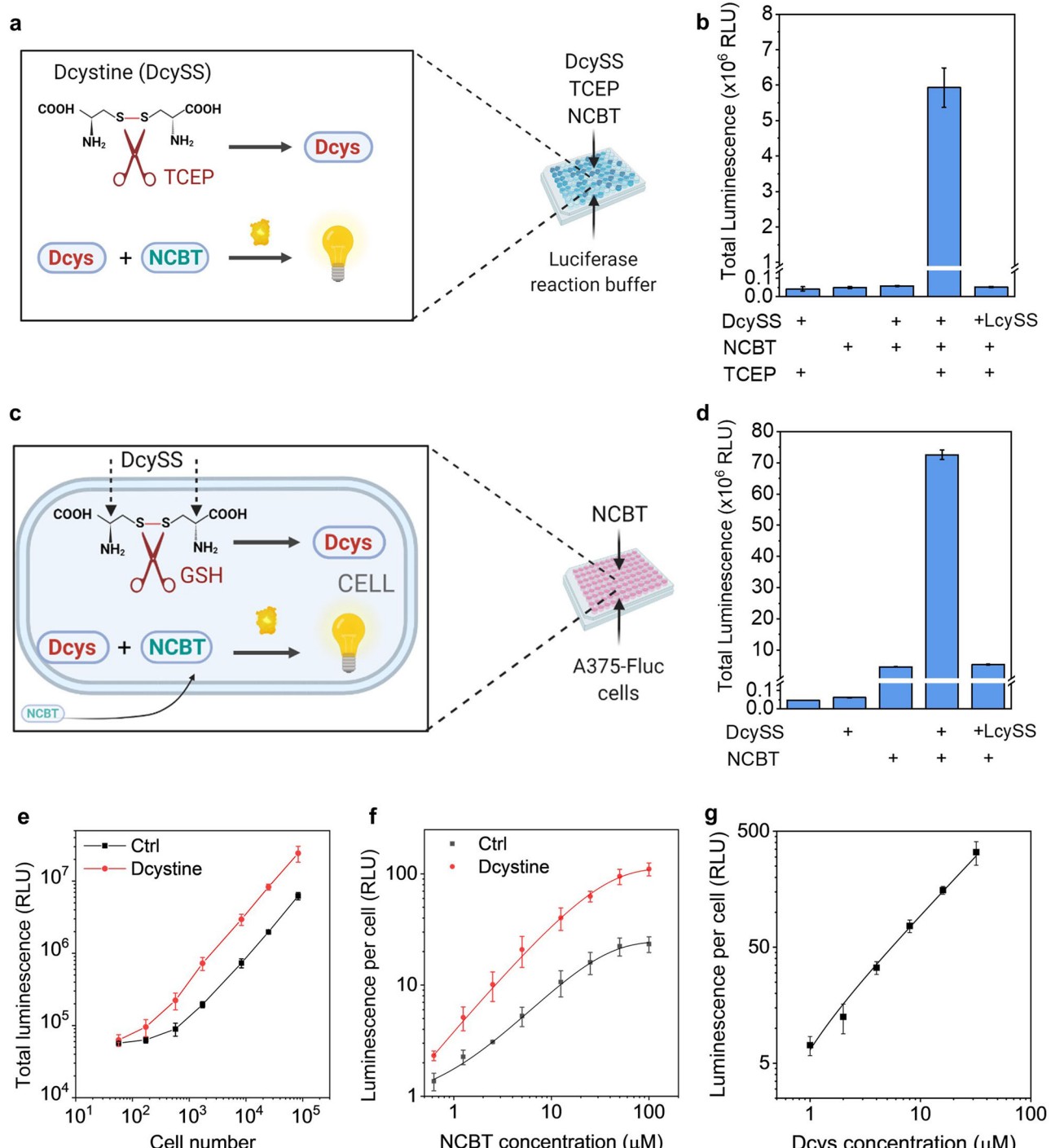

**Fig. 2 | Performance of BioLure assay in the cell-free reaction buffer and on live luciferase-expressing cell lines. a** General scheme of BioLure assay in cell-free reaction buffers. DcySS D-cystine, dimer of D-cysteine (Dcys), LcySS L-cystine, dimer of L-cysteine, TCEP tris(2-carboxyethyl)phosphine, NCBT 6-amino-2-cyanobenzothiazole. Created with BioRender.com. **b** Total bioluminescence output (from 0 to 30 min) of the bioorthogonal reaction in the reaction buffer, at the presence/absence of key reactants. **c** General scheme of BioLure assay in A375-Fluc-

eGFP cells. Created with BioRender.com. **d** Total bioluminescence output (from 0 to 30 min) of the bioorthogonal reaction in A375-Fluc-cells at the presence/absence of key reactants. Dependence of luminescence output on (**e**) number of cells used, **f** amount of NCBT added to the assay, and (**g**) amount of Dcys added to the cells during the electroporation. In **e**, **f**, Ctrl means cell samples with NCBT but without DcySS. Data are presented as the mean ± s.d. (*n* = 3).

as control cells (electroporated without any polymer). The luminescence signal of Dex-Dcys quickly peaked after 5 min upon the addition of NCBT and gradually decreased back to the background level within 30 min. Furthermore, when adding different amount of Dex-Dcys in the electroporated sample, and the results in Fig. 3d show that the MFI and the total luminescence signal of the electroporated samples had exactly the same trend.

Based on the results shown in Fig. 3d, we estimated the amount of Dex-Dcys delivered in the cells by the correlation curve in Fig. 2g. When using 2.5–10 μg Dex-Dcys in the electroporation, 0.86–1.06‰ of the Dex-Dcys was delivered (Fig. S7). However, a further increase in Dex-Dcys amount in the electroporation to 20 μg did not lead to more MOIs in the cells. The results calculated by bioluminescence output was further

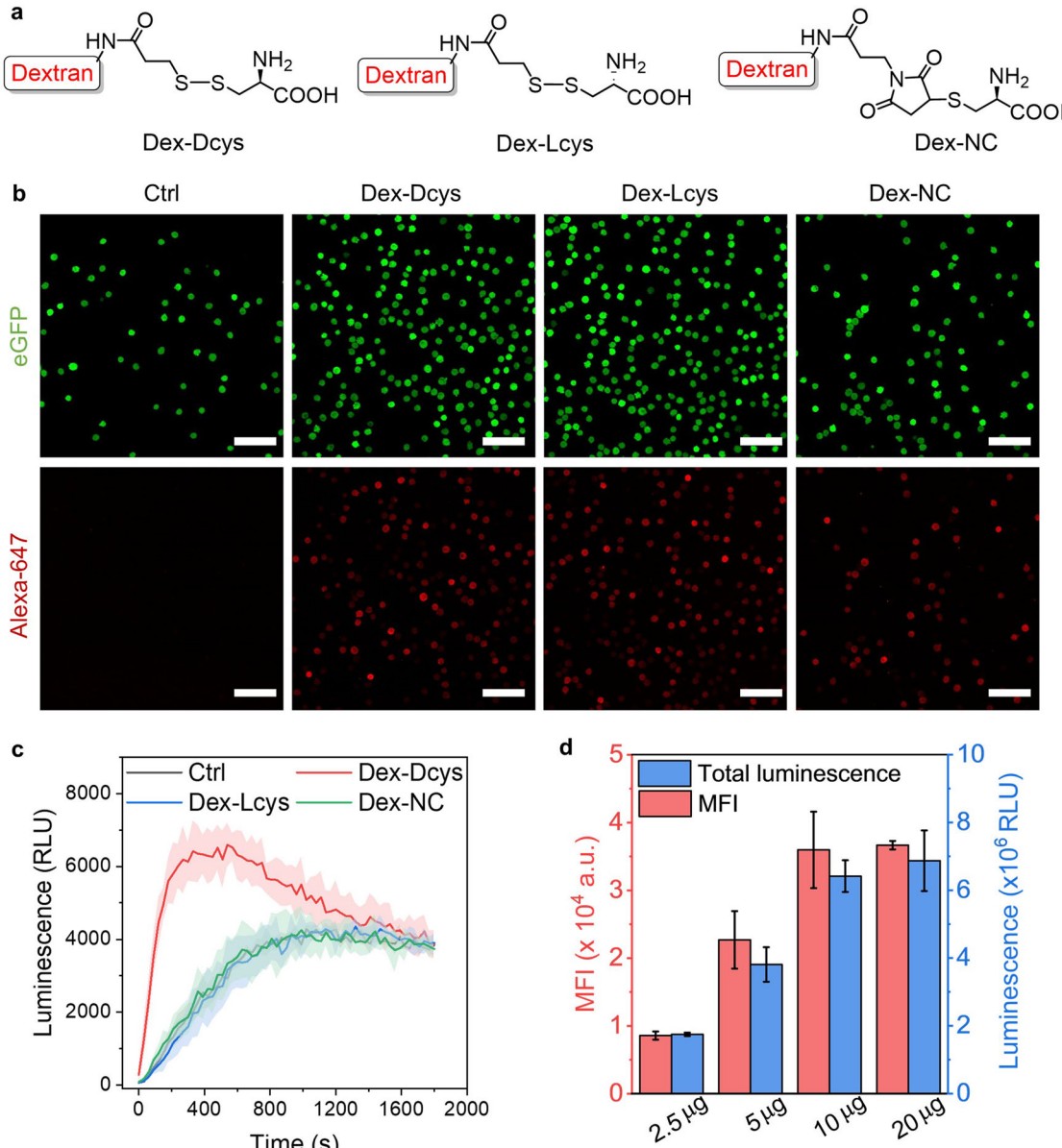

**Fig. 3 | Intracellular delivery and quantification of Dcys-labelled dextran polymers. a** Chemical structures of D-cysteine-modified dextran polymers. **b** The confocal microscopy images of A375-Fluc-eGFP cells right after the electroporation of Alexa647 labelled dextran polymers. Ctrl means electroporated cells without any polymer. Scale bar: 100 μm. **c** Real-time luminescence signal of electroporated cells after the addition of NCBT. The error bars are presented as filled areas. **d** The mean fluorescence intensity (MFI) and total luminescence of cells electroporated with different amount of Dex-Dcys (2.5–20 μg). Data are presented as the mean ± s.d. ($n = 3$).

validated by a complementary quantitative flow cytometry method using the standard curve set by fluorescence calibration Quantum™ beads[51]. The results showed consistency with bioluminescent estimation (Fig. S7). The maximum dextran delivered per cell ($5 \times 10^{-14}$ g) equals *ca.* 3 million dextran molecules per cell, resulting in an intracellular dextran concentration of 2.2 μM. This is a reasonable concentration compared with other reported electroporation-induced dextran delivery, despite using different devices and settings[52]. However, due to the heterogeneous nature of the bulk electroporation technique[53], the delivery efficiency varied on the individual cell level, evidenced by a wide distribution of fluorescence intensity within the electroporated cell populations (Fig. S8). This means the calculated delivered dextran amount is an average estimation of all electroporated cells. In summary, the results of Dex-Dcys show that the labeling of Dcys had minimal changes in MOI's structure, and the MOIs could be successfully delivered to the cytoplasm by electroporation with detectable bioluminescence output. The

bioluminescence also showed a good correlation with complementary verification methods.

Following the case study using dextran, we investigated a more complex MOI, lysozyme. We first labeled the Dcys tag on lysosome using a similar synthetic strategy in the Dex-Dcys preparation but failed due to protein crosslinking by SPDP linker. Instead, we designed a two-step reaction (Fig. 4a), first converting amine residuals on lysozyme to thiol by Traut's reagent, and then reacting with a home-made linker (S-(pyridin-2-ylthio)-D-cysteine, Py-Dcys, NMR spectrum in Fig. S9). The Dcys-conjugated lysozyme (Dlyso) was characterized by gel electrophoresis to identify the structural variations after conjugation. On the native-PAGE gel (Fig. S10), Dlyso showed reduced mobility and a tail compared with the original protein, probably due to the change of charges after the conjugation of Dcys. After we extracted the proteins from the Native-PAGE gel, lysozyme and Dlyso were subjected to TCEP reduction to release the Dcys tag for BioLure assay quantification, and the luminescence output was recorded

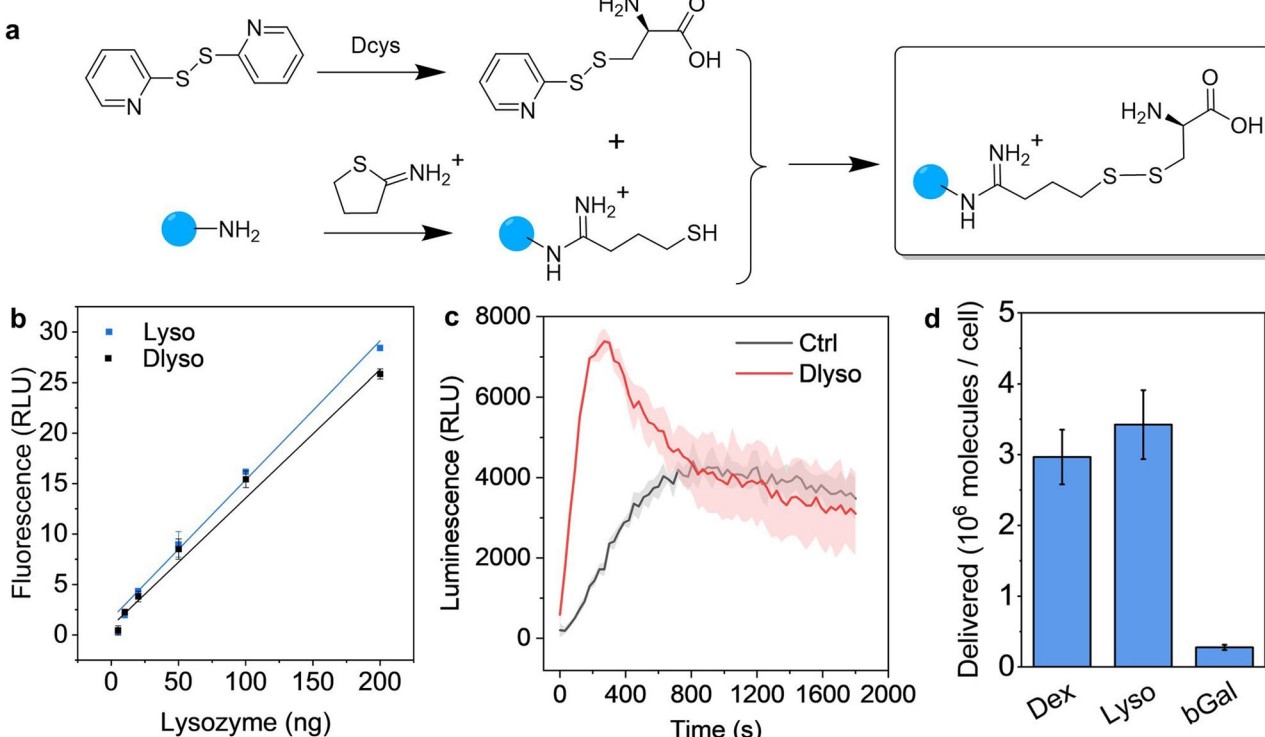

**Fig. 4 | The preparation, characterization, and intracellular delivery quantification of Dcys-labelled proteins. a** Synthetic scheme of lysozyme labeled by Dcys tag (Dlyso). The blue dot represents proteins with amine residuals. **b** Calibration curves of lysozyme and Dlyso using the EnzChek™ Lysozyme Assay. **c** Real-time luminescence signal of electroporated A375-Fluc-eGFP cells with Dlyso and without Dlyso (Ctrl). **d** The average number of delivered molecules per cell, when electroporated with 20 μg proteins. Data are presented as the mean ± s.d. (filled area, $n = 3$).

in cell-free reaction buffers similar to Fig. 2a. As shown in Fig. S10, Dlyso, after TCEP reduction, had a significantly higher signal than background and controls (Dlyso without TECP reduction), suggesting the Dcys tag was successfully conjugated and cleaved upon treatment with reducing reagents. We further characterized the structural and functional variation of Dlyso by reducing SDS-PEG gel electrophoresis and a lysozyme activity assay. After protein denature and reduction, Dlyso showed identical bands on the gel (Fig. S11), suggesting no non-cleavable agglomeration in Dlyso. Regarding the enzymatic function of lyzozyme, the activity was well retained after conjugation, with almost identical calibration curves before and after conjugation (Fig. 4b).

We used Dlyso in the electroporation-induced intracellular delivery. As shown in Fig. 4c, the real-time luminescence signal was similar to Dex-Dcys. The signal quickly peaked after 5 min and decreased to the background level. We estimated the total amount of Dlyso based on the luminescence signal and the correlation curve (Fig. 2g). As shown in Fig. 4d, there were *ca.* 3.4 million lysozymes delivered per cell on average. This is slightly larger than the amount of dextran intracellular delivered when using the same mass (20 μg) in the electroporation. Considering the dextran used in this study and lysozyme have rather similar molecular weights (10 and 14 kDa, respectively), the higher intracellular delivery efficiency of lysozyme was probably due to its positive charge facilitating the electrophoretic transport of proteins in the cytoplasm[54].

Regarding the applicability of BioLure on even larger biomolecules, we investigated another protein payload, β-Galactosidase (bGal, 465 kDa). First, we conjugated Dcys on bGal and then characterized the Dcys-labelled bGal (DbGal) by gel electrophoresis. Due to the complex structure, we did not manage to separate the protein on Native-Page gel despite trying different gel compositions and running voltages. But in the reducing SDS-PAGE gel, we managed to get clear bands of both bGal and DbGal (Fig. S11). The original bGal showed multiple bands, but the main band was 125 kDa, corresponding to one subunit of the protein. Similarly, DbGal also showed

multiple bands with the main band at the same molecular weight. This means after reduction, DbGal could be restored to its original form.

Then we tested DbGal in the electroporation-mediated intracellular delivery. As shown in Fig. S12, the real-time luminescence signal output followed a similar trend as Dcys-labelled dextran and lysozymes. Compared with lysozymes and dextran, the final estimated delivered DbGal was lower, *c.a.*, 0.28 ± 0.04 million per cell (Fig. 4d). The lower delivery efficiency could be attributed to the large size of bGal, but other factors such as charge, structure, and membrane association may also affect the electroporation-mediated cell entry[54–56].

Furthermore, we verified the Dlyso and DbGal intracellular delivery results by complementary methods (fluorescence enzymatic assays). For lysozyme, we used EnzChek™ Lysozyme Assay and analyzed Dlyso in the cell lysate after electroporation. For β-Galactosidase, we used FACS Blue LacZ beta Galactosidase Detection Kit, which allows for quantifying the protein amount in intact cells. We obtained similar delivery results from enzymatic activity assays (2.8 ± 0.3 million per cell for lysozyme, and 0.32 ± 0.04 million per cell for bGal), compared with BioLure results in Fig. 4d. Thus, it is concluded that BioLure assay could be applied to quantify the delivery of protein MOIs of various sizes.

To explore the applicability of BioLure assay on different cells, we chose HEK 293 cells which were transfected and transiently expressed luciferase (Fig. S13). The aim is to explore if BioLure is still applicable at a lower cellular luciferase expression level. If so, anyone who wants to use this assay does not need to spend weeks establishing a stable cell line but simply use a plasmid or mRNA to express luciferase transiently in the specific cell of interest before intracellular delivery. Although transiently transfected cells only express luciferase in a limited time, the transfection process is much easier and faster and still provides a sufficient time window for intracellular delivery studies.

In this study, HEK 293 cells were transfected by a plasmid (co-expression of luciferase and eGFP). The transfection efficiency was confirmed by flow cytometry (Fig. S13). More than 80% positive events were

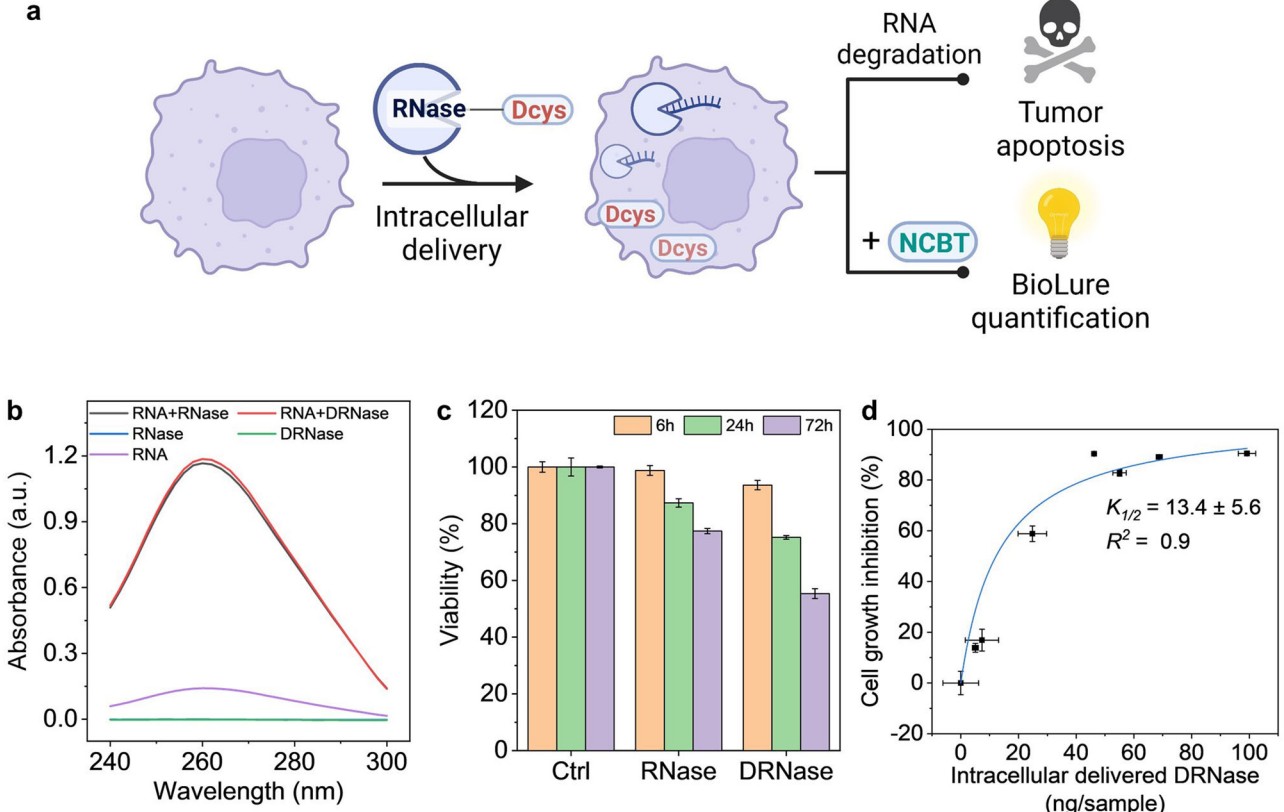

**Fig. 5 | The preparation, characterization, and intracellular delivery quantification of Dcys-labelled RNase (DRNase). a** Scheme of intracellular DRNase quantification and tumoricidal activity. Created with BioRender.com. **b** UV-absorbance spectra of yeast RNA (purple), RNase (blue), DRNase (green), and yeast RNA digested by RNase (black) or DRNase (red). **c** Cell viability after 6, 24, and 72 h post-electroporation with RNase or DRNase. Ctrl means electroporated cells without any enzyme. **d** Correlation between intracellular DRNase, and cell growth inhibition after 72 h electroporation. The data were presented as the mean ± s.d. ($n = 4$), fit by the Michaelis-Menten equation.

achieved after 24 h transfection and maintained after 48 h, suggesting a high transfection efficiency. The luciferase expression was also confirmed by its substrate (Fig. S13). The luminescence of luciferase-expressed HEK293 cells after Dlyso intracellular delivery is shown in Fig. S14. Using the same amount of cells, transfected HEK293 had a much lower signal in the Dlyso sample compared with A375-Fluc-eGFP and a lower background signal in the negative control, because of a lower luciferase expression level. However, the trend of the real-time luminescence output in both cell lines was similar, with an instant signal increase within 5 min of NCBT addition followed by a slow decay. These results suggest BioLure is applicable to cells with both stable and transient luciferase-expressing cells.

Finally, we explored BioLure's capability to evaluate protein MOIs with therapeutic relevance. We chose RNase A because it is an endoribonuclease that specifically degrades single-stranded RNA and exhibits tumoricidal activity (Fig. 5a)[57,58]. Despite the previous reports of intracellular RNase A delivery nanoformulations for cancer cell elimination, the cytosolic concentration of RNase A needed to achieve satisfactory therapeutic efficacy is still unknown. We believe this fundamental insight would be valuable for designing protein intracellular delivery formulations.

Dcys-labelled RNase A (DRNase) was synthesized using the strategy displayed in Fig. 4a. The structure and function of DRNase were investigated using gel electrophoresis and enzymatic assays. Similar to Dlyso, DRNase did not show changes in protein molecular weight according to the reducing SDS-PAGE gel (Fig. S11) after conjugation. The RNA degradation capability of DRNase was also identical to the original RNase A, proved by the increased UV absorbance at 260 nm after yeast RNA digestion (Fig. 5b). The RNA degradation capability of both RNase A and DRNase could be inhibited by recombinant ribonuclease inhibitors (Fig. S15), suggesting

the Dcys conjugation did not have a major impact on either the ribonuclease activity or the specific binding between RNase and its inhibitor.

Next, we investigated the cytotoxic effects of RNase and DRNase on A375-Fluc-eGFP melanoma cells. As shown in Fig. S16, without intracellular delivery, the cell viability was higher than 90% for both RNase and DRNase treated samples even after 72 h. With electroporation-induced intracellular delivery, both RNase and DRNase treated cells already showed a reduction in relative cell viability 24 h post-electroporation (Fig. 5c). After 72 h, the viability of cells with intracellular DRNase further dropped to 55%, suggesting a prominent growth inhibitory effect. Then, we delivered different amounts of DRNase into cells, by varying the amount of DRNase in the electroporation (0–20 μg). We quantified the intracellular DRNase amount by BioLure and measured the cell viability 72 h post-electroporation. As shown in Fig. 5d, the concentration-dependent growth inhibition effect of DRNase was fit via a Michaelis-Menten model with an R-square of 0.9. The half-inhibition concentration is 13.4 ± 5.6 ng per sample (200,000 cells), equal to an estimated 2.8 ± 1.2 million of DRNase molecules per cell. This quantitative estimation suggests that, in the case of RNase A, at least millions of therapeutic proteins need to be delivered to the cytoplasm to degrade the intracellular RNA and achieve sufficient growth inhibition in A375 melanoma cells. Despite our efforts to ensure consistency, we admit that the standard deviation of the half-inhibition values was notably large. This variability can be attributed to several reasons including a limited number of data points, variable electroporation efficiency from sample to sample, and the non-ideal fitting algorithm. Such limitations of our method could be overcome by increasing the number of data points and refining the fitting method in future work.

## Conclusions

In summary, we report an unconventional strategy (BioLure assay) for intracellular delivery quantification via bioorthogonal luminescent reactions. The assay requires minimal labeling on the MOIs, which does not interfere with the structure and function of the model proteins used in this study. The quantification procedure is simple and fast, with an instant report from highly efficient and specific reactions. Furthermore, we demonstrated that this analytical toolset is sensitive and applicable to protein MOIs with complex structures and properties. Specifically, this strategy could be used on therapeutically relevant cargo and cell types. By unleashing the potential of bioorthogonal reaction in live-cell intracellular delivery quantification, we predict BioLure will address the great need in the field, and establish a quantitative link between available cytosolic MOIs after intracellular delivery with their therapeutic outcomes.

## Methods

### Materials and reagents

Methanol, diethyl ether, Traut's Reagent (2-iminothiolane hydrochloride), lysozyme from chicken egg white (~70000 U mg$^{-1}$), β-Galactosidase from *Escherichia coli* (lyophilized, powder, ~140 U mg$^{-1}$), 4-(2-hydroxyethyl)-1-piperazineethanesulfonic acid (HEPES), phosphate buffered saline (PBS), ethylenediaminetetraacetic acid (EDTA), D-cystine (DcySS), 6'-amino-D-luciferin (D-amLu), 6-amino-2-cyanobenzothiazole (NCBT), adenosine 5'-triphosphate disodium salt hydrate (ATP), magnesium chloride hexahydrate (MgCl$_2$), recombinant luciferase from *Photinus pyralis* (firefly), anhydrous dimethyl sulfoxide (DMSO) and DMSO-$d_6$ were purchased from Sigma-Aldrich (St. Louis, MO, USA). Dulbecco's modified Eagle's medium (DMEM), geneticin and fetal bovine serum (FBS) were purchased from Life Technologies Gibco, USA. Amino-dextran (10 kDa, 5.1 mole amine per mole of dextran), succinimidyl 3-(2-pyridyldithio)propionate (SPDP), Alexa Fluor™ 647 NHS Ester (Succinimidyl Ester) were purchased from Thermo Fisher Scientific, USA. D-cysteine hydrochloride monohydrate (Dcys), L-cysteine hydrochloride monohydrate (Lcys), L-cystine dihydrochloride (LcySS), 2, 2'-dithiodipyridine, tris(2-carboxyethyl)phosphine hydrochloride (TCEP), and N-succinimidyl 3-maleimidopropinate (BMPS) were purchased from TCI Europe N.V. (Belgium).

The following chemicals were obtained as indicated: Puromycin (Peprotech, USA), CellTiter-Glo® Luminescent Cell Viability Assay (Promega, USA), GSH/GSSG-Glo™ Assay (Promega, USA), Recombinant RNasin® Ribonuclease Inhibitor (Promega, USA), RNase A from bovine pancreas (Roche CustomBiotech, Germany), RNA from yeast (Roche CustomBiotech, Germany), Hank's Balanced Salt Solution (HBSS, Hyclone, USA), non-essential amino acids (NEAA, HyClone, USA), penicillin-streptomycin (HyClone, USA), L-glutamine (HyClone SpA, USA), and trypsin (HyClone SpA, USA).

### BioLure assay in the cell-free reaction buffer

BioLure assay in the cell-free reaction buffer were adapted from the protocol reported in the literature with minor modifications[33]. The cell-free reaction buffer was based HBSS−HEPES buffer at pH 7.4, supplemented by recombinant firefly luciferase (10 μg mL$^{-1}$), ATP (1 mM), and MgCl$_2$ (5 mM). Then DcySS or LcySS (final concentration at 50 μM) were added to the reaction buffer, with or without TCEP (final concentration at 100 μM). NCBT (5 mM stock solution in DMSO) was added right before the luminescence detection at a final concentration at 50 μM in the reaction buffer. The luminescence was recorded at 25 °C using Varioskan™ LUX multimode microplate reader (Thermo Fisher Scientific Inc., USA). The measurement time was set at 1000 ms, and the recording lasted for 30 min at a kinetic interval of 30 s. The total luminescence was calculated based on the integral of the real-time luminescence during the 30 min.

### Cell culture

A375-eGFP-Fluc cells (A375-Fluc-Neo/eGFP-Puro, provided by Imanis Life Sciences, USA) were cultured in DMEM with 4.5 g L$^{-1}$ glucose, supplemented with 10% of FBS, 1% of L-glutamine, 1% of NEAA, penicillin (100 IU mL$^{-1}$), streptomycin (100 μg mL$^{-1}$), geneticin (0.6 mg mL$^{-1}$) and puromycin (1 μg mL$^{-1}$). HEK293 cells (ATCC) were cultured in DMEM with 4.5 g L$^{-1}$ glucose, supplemented with 10% of FBS, 1% of L-glutamine, 1% of NEAA, penicillin (100 IU mL$^{-1}$) and streptomycin (100 μg mL$^{-1}$). The cells were cultured in the 5% CO$_2$-incubator at 37 °C, and 95% relative humidity. The culture media were changed every 2-3 days, and the cells were passaged at ~90% confluence using 0.25% (v/v) trypsin EDTA/PBS.

### Electroporation-mediated intracellular delivery

For a typical electroporation process, $4 \times 10^5$ cells were resuspended in Lonza SF 4D-Nucleofector™ Solution (16.4 μL Nucleofector™ Solution + 3.6 μL Supplement from SF Cell Line 4D-Nucleofector™ X Kit S, Lonza Bioscience, Switzerland), followed by the addition of 2 μL molecules of interests (DcySS, LcySS, Dex-Dcys, Dex-Lcys, Dex-NC, Lyso, Dlyso, RNase, or DRNase, stock solution in MilliQ water). For negative control, only 2 μL MilliQ water was added. The mixture was transferred to one well of Nucleocuvette™ Strip carefully without air bubbles. Then, the Nucleocuvette™ Strip was electroporated with 4D-Nucleofector® X Unit using the pre-optimized program (Code: FF-120 for A375-eGFP-Fluc cells). After electroporation, the Nucleocuvette™ Strip was incubated in a cell incubator (37 °C, 5% CO$_2$ and 95% relative humidity) for 10 min for the cell recovery. Then, 80 μL pre-warmed cell culture medium was added to resuspend the cells in the well, and the sample was transferred to an Eppendorf and centrifuged at 90 g for 10 min. The supernatant was carefully removed without disturbing the cell pellet. Then the cell pellet was resuspended in HBSS-HEPES buffer (pH 7.4) and added to 96-well microplates (white wall with transparent bottom) for BioLure assay analysis.

### BioLure assay of electroporated cells

BioLure assay of electroporated cells was performed after electroporation-mediated intracellular delivery. NCBT (5 mM stock solution in DMSO) was added to the cell suspension in HBSS-HEPES buffer, and electroporated cells with the same amount of DMSO but without NCBT was used as the negative control. The luminescence was recorded using Varioskan™ LUX multimode microplate reader at 25 °C. The measurement time was set at 1000 ms, and the recording lasted for 30 min at a kinetic interval of 30 s. The total luminescence was calculated based on the integral of the real-time luminescence during the 30 min.

### Synthesis and characterizations of Dex-Dcys, Dex-Lcys and Dex-NC

**Synthesis of Dex-Dcys and Dex-Lcys.** Dex-Dcys and Dex-Lcys were synthesized by a two-step reaction. First, amino-dextran (4 mg, 0.4 μmol) was conjugated with SPDP linker (0.6 mg, 1.96 μmol) in 1 mL PBS−EDTA buffer (pH 8.0) for 0.5 h at room temperature. Then, the unreacted SPDP was removed by centrifugation at 16,110 ×g for 10 min using Amicon Ultra-0.5 Centrifugal Filter Unit (3 kDa MWCO). The SPDP conjugated dextran (Dex-SPDP) was further purified by washing with PBS-EDTA buffer 3 times using Amicon Ultra-0.5 Centrifugal Filter Unit (3 kDa MWCO). Then the final products were either mixed with L-cysteine hydrochloride monohydrate (0.7 mg, 4 μmol) or D-cysteine hydrochloride monohydrate (0.7 mg, 4 μmol) in PBS−EDTA buffer. The UV-absorbance at 343 nm was measured before and at 15 min after the addition of cysteine, to estimate the number of cysteine conjugated per dextran molecule according to the method reported in literature[59]. After 0.5 h, the reaction mixture was again purified by centrifugation at 16,110 ×g for 10 min using Amicon Ultra-0.5 Centrifugal Filter Unit (3 kDa MWCO). The products (Dex-Dcys and Dex-Lcys) were washed 6 times by MilliQ water and freeze-dried overnight. The lyophilized dextran polymers were stored at −20 °C until use.

**Synthesis of Dex-NC.** First, amino-dextran (4 mg, 0.4 μmol) was conjugated with BMPS linker (0.52 mg, 1.96 μmol, in 1 mL PBS-EDTA buffer (pH 8.0) for 0.5 h at room temperature. The intermediate product (Dex-mal) was purified similarly as Dex-SPDP using Amicon Ultra-0.5

Centrifugal Filter Unit (3 kDa MWCO). Then, Dex-mal was subjected to reaction with D-cysteine hydrochloride monohydrate (0.7 mg, 4 µmol) in 1 mL PBS−EDTA buffer for 0.5 h at room temperature, and purified similarly as Dex-Dcys. The final product (Dex-NC) was freeze-dried and stored at −20 °C until use.

**Fluorescence labelling of Dex-Dcys, Dex-Lcys and Dex-NC.** The fluorescent derivatives of Dex-Dcys, Dex-Lcys and Dex-NC were synthesized by reacting with Alexa Fluor™ 647 NHS Ester. Briefly, 2 mg Dex-Dcys, Dex-Lcys or Dex-NC was dissolved in 1 mL PBS−EDTA buffer (pH 8.0) and then Alexa Fluor™ 647 NHS Ester (0.02 µmol, 0.4 mM stock in DMSO) was added. The reaction was allowed at room temperature for 0.5 h with protection from light. The unreacted Alexa Fluor™ 647 was removed by washing with PBS-EDTA buffer 3 times using Amicon Ultra-0.5 Centrifugal Filter Unit (3 kDa MWCO), and then further washed by MilliQ water 3 times before lyophilization.

**Characterizations of polymers.** The synthesized polymers, inducing intermediate products (Dex-SPDP and Dex-mal) were characterized by $^1$H-NMR on a Bruker Vertex 70 spectrometer to confirm the structure. The results were analyzed by MestreNova software and plotted in Figs. S3 and S4 in the supporting information.

The polymers were also subjected to elemental analysis with an automatic elemental analyzer vario MICRO cube (HANAU Elementar Analysensysteme GmbH, Germany, Serial no. 15082023). Sulfanilamide standard (>99.9%, Elementar, Germany) was used as the standard. Analyzes were performed in CHNS mode ($O_2$ dosing time: 70 s; Autozero delay: 10 s; Peak anticipation: N 50 s, C 120 s, H 100 s, S 70 s). Carbon was determined as $CO_2$, hydrogen as $H_2O$, nitrogen as $N_2$ and sulfur as $SO_2$. $N_2$ is not adsorbed in the adsorption column and is the first measuring component to enter the thermal conductivity detector (TCD). $CO_2$, $H_2O$ and $SO_2$ are adsorbed together in the adsorption column. The adsorption column is heated stepwise to desorption temperatures of $CO_2$ (60 °C), $H_2O$ (140 °C) and $SO_2$ (210 °C). The measured gas enters the detector with the carrier gas (He) one by one. The percentage elemental concentration of the element in the sample is calculated using formula $c\% = \frac{a \cdot 100 \cdot f}{w}$, where $c$ is the element concentration [%], $a$ absolute element content [mg], $f$ the daily factor and $w$ the sample weight [mg].

Fluorescently labeled Dex-Dcys, Dex-Lcys and Dex-NC were further characterized by size exclusion chromatography (SEC). The system consists of a Waters 515 HPLC pump, Biotech DEGASi GPC Degasser, Waters 717 plus Autosampler, Shimadzu RF535 Fluorescence HPLC monitor and Waters 2410 Differential Refractometer together with Waters Ultrahydogel 120, 250, 2000 7.8 × 300 mm columns and a guard column. The column was kept at 30 °C. Fluorescence excitation was set to 650 nm and emission 675 nm. Pullulan standards by Polymer Standard Service were used for the molecular weight calibration.

## Confocal microscopy imaging
Cells electroporated with fluorescently labeled dextran derivatives were suspended in HBSS−HEPES buffer and then sediment in a 35 mm glass bottom dish before confocal imaging. Then the cells were imaged by a Leica Stellaris 8 confocal microscope (Leica Microsystems, Wetzlar, Germany). The images acquired were processed by Fiji 1.51 software.

## Flow cytometry analysis
All the flow cytometry analysis was performed on BD LSR-II Cell Analyzer flow cytometer, and the data was processed by FlowJo™ software. Regarding the quantitative flow cytometry analysis, Quantum™ Alexa Fluor® 647 Molecules of Soluble Fluorochrome (MESF) beads (Bangs Laboratories, Inc, USA) were used. Basically, 5 beads with pre-determined Alexa Fluor® 647 labelled molecules were run on the same day same by the same flow cytometer at the same setting with flow cytometry cell samples. The mean fluorescence intensity (MFI) of each bead (from triplicate samples) were calibrated using the QuickCal® analysis template as instructed by

the manufacturer. BD FACSDiva™ logarithmic regression was selected to fit the data.

## Synthesis of Py-Dcys
Py-Dcys (S-(pyridin-2-ylthio)-D-cysteine) was synthesized by reacting 2, 2'-dithiodipyridine with D-cysteine hydrochloride following literature procedures[60]. Briefly, 2, 2'-dithiodipyridine (0.22 g, 1 mmol) was dissolved in 2.5 mL methanol with 0.1 mL acetic acid. Then the second reactant (Dcys HCl, 78.81 mg, 0.5 mmol), pre-dissolved in 1 mL methanol, was slowly added. The reaction was left stirring at room temperature overnight. On the following day, the reaction mixture was precipitated by cold diethyl ether (40 mL), and centrifuge at 6000 rpm for 5 mins to remove the supernatant. The precipitation was redissolved in 2 mL methanol, and precipitated in cold diethyl ether (20 mL) again. After removal of the supernatant, the precipitation was repeated for another time. Finally, the white precipitate was collected and dried overnight to remove residual diethyl ether. The synthesized Py-Dcys was characterized by $^1$H-NMR on a Bruker Vertex 70 spectrometer to confirm the structure.

## Protein labelling and characterizations
Lysozyme and RNase A were labelled via two-step reactions. First, 5 mg protein was dissolved in 1 mL PBS−EDTA buffer (pH 8). Then, 0.5 mg Traut's Reagent (2-iminothiolane hydrochloride, 3.6 µmol) in 50 µL PBS−EDTA buffer (pH 8.0) was added, and stirred at room temperature for 30 min. The proteins were purified by Amicon Ultra-0.5 Centrifugal Filter Unit (3 kDa MWCO), by washing with PBS−EDTA buffer for 4 times. During each washing step, the sample was centrifuged at 16,110 g for 10 min. The purified proteins were diluted in PBS-EDTA buffer to make the volume to 1 mL, followed by the addition of Py-Dcys (1.1 mg, 4.2 µmol). The UV-absorbance at 343 nm was measured before and at 15 min after the addition of Py-Dcys, to estimate the number of cysteine conjugated (on average 1.1 Dcys per lysozyme, and 0.7 Dcys per RNase A). The reaction was allowed at room temperature for 30 min, and the proteins were purified again by Amicon Ultra-0.5 Centrifugal Filter Unit (3 kDa MWCO) using the same centrifugation conditions. The proteins were washed 6 times by MilliQ water instead of PBS−EDTA buffer, and freeze-dried overnight. The lyophilized proteins were stored at −20 °C until use.

β-Galactosidase (bGal) were labelled similarly as Dex-Dcys via two-step reactions. First, bGal (1 mg) was conjugated with SPDP linker (0.22 mg, 0.7 µmol) in 1 mL PBS−EDTA buffer (pH 8.0) for 0.5 h at room temperature. The protein was purified by washing with PBS-EDTA buffer 3 times using Amicon Ultra-0.5 Centrifugal Filter Unit (10 kDa MWCO, 16,110 g for 10 min each time). Then D-cysteine hydrochloride monohydrate (0.22 mg, 1.4 µmol) in PBS−EDTA buffer was added and the UV-absorbance at 343 nm was measured before and at 15 min after the addition of cysteine, to estimate the number of D-cysteine conjugated (11.3 Dcys on each bGal subunit, 116 kDa). After 0.5 h, the reaction mixture was again purified by centrifugation at 16,110 × $g$ for 10 min using Amicon Ultra-0.5 Centrifugal Filter Unit (10 kDa MWCO). The purified D-cysteine conjugated bGal (DbGal) was freeze-dried overnight and stored at −20 °C until use.

The proteins were characterized by gel electrophoresis on SDS-PAGE gel and Native PAGE gel. For SDS−PAGE Gel, proteins were mixed with Novex™ Tris-Glycine SDS Sample Buffer (Thermo Fisher, USA) and reduced by TCEP or β-mercaptoethanol, and heated at 95 °C for 10 min. The samples were loaded on 4–15% Mini-PROTEAN® TGX™ Precast Protein Gels along with Thermo Scientific Spectra Multicolor Broad Range Protein Ladder (10–260 kDa). Then electrophoresis was performed at 160 V for 30 min. The gel was stained by PageBlue™ protein staining solution (Thermo Fisher, USA), and imaged by GelDoc™ Imaging System (BioRad, USA).

For Native-PAGE gel electrophoresis, a Tris-glycine polyacrylamide gel system consisting of a 4.5% stacking gel and a 6% separation gel was used to separate 20 µg purified protein under nondenaturing conditions in a Mini-PROTEAN electrophoresis chamber (BioRad) in running buffer

(25 mM Tris, 192 mM glycine, pH ~8.0) at 30 V overnight at 4 °C. Further gels were washed 3 times in ddH2O for 5 minutes, incubated in 10 mL of InstantBlue® Coomassie Protein Stain (ab119211) for 1 h, distained twice in 100 mL of ddH2O for 15 min. After staining, the protein bands were cut from the gel into small pieces, and the proteins were extracted in to HBSS buffer by gentle shaking at 4 °C overnight. Then, the gel pieces were removed by centrifugation, and the protein-containing supernatants were collected and washed 3 times by Amicon Ultra-0.5 Centrifugal Filter Unit (3 kDa MWCO), to remove salts and impurities diffusing from the gel during the extraction. The concentrated and purified proteins were diluted in the reaction buffers and the luminescence was recorded according to the procedures previously described in the *BioLure assay in the cell-free reaction buffer* section.

### Lysozyme activity evaluations

The enzymatic activity of lysozymes before and after Dcys modification was evaluated by Invitrogen™ EnzChek™ Lysozyme Assay Kit (Thermo Fisher, USA), following the manufacture's protocol. Lysozymes of different concentrations (0–200 ng per well) were diluted in 1X reaction buffer and a volume of 50 μL was added to one well in a 96-wellplate, and 50 μL lysozyme substrate working solution (50 μg mL$^{-1}$ substrate in 1X reaction buffer) was added. The plate was incubated for 30 min at 37 °C protected from light, and the mean fluorescence (RLU) was measured by Varioskan™ LUX multimode microplate reader. The excitation wavelength was 494 nm and the emission was 518 nm.

Regarding lysozyme quantification in the cell lysate, 100,000 electroporated cells with Dlyso were washed with cold PBS, and lyzed on ice for 10 min with 100 μL lysis buffer (containing protease inhibitor cocktail). The sample was centrifuged at $12,000 \times g$ for 10 min at 4 °C, and the supernatant was collected and analyzed by EnzChek™ Lysozyme Assay Kit following the protocol described above. A control sample consisted of electroporated cell lysate without Dlyso was also measured using the same assay, to identify the endogenous lysozyme amount. Finally, the intracellularly delivered lysozyme was calculated by subtracting negative control from the Dlyso electroporated sample.

### RNase activity evaluations

The enzymatic activity of RNase was evaluated by a well-established protocol from literature[61] with modifications. First, prepare crude yeast RNA stock solution at 1 mg mL$^{-1}$ in 62.5 mM Tris-buffer (pH 8.0). Then 40 μL of diluted RNase or DRNase solution was added to 160 μL RNA stock; the mixture was heated at 37° for 10 min. The reaction was stopped by the addition of 200 μL of 6% HClO$_4$ (Perchloric acid). After 5 min on ice, the samples were centrifuged at $16,110 \times g$ for 3 min to precipitate the undegraded RNA and proteins. Then 100 μL of the supernatant was diluted with 500 μL Milli-Q water, and the UV absorbance spectra from 240 to 300 nm was measured.

The RNase activity before and after Dcys labelling was further evaluated by RNA gel electrophoresis. The 16 S and 23 S rRNA from *E.coli* was used as the substrate for RNase. For each sample, 1 ug rRNA in 10 μL TE buffer was mixed with RNase or DRNase (0.04, 0.4, 4 and 40 ng) with 40 U Recombinant RNasin Ribonuclease Inhibitor (Promega, USA). The mixture was heated at 37° for 10 min and loaded on 2% agarose gel, run at 80 V for 2 h. Gel was imaged with ChemiDoc™ Imaging System (Bio-Rad).

### β-Galactosidase activity evaluations

The enzymatic activity of β-Galactosidase before and after Dcys modification was evaluated by FACS Blue LacZ beta Galactosidase detection kit (Abcam, UK), following the manufacture's protocol. β-Galactosidase of different concentrations (0–25 ng per well) were diluted in 1X reaction buffer and a volume of 50 μL was added to one well in a 96-wellplate, followed by the addition of 100 μL reaction buffer and 50 μL substrate reagent. The plate was incubated for 20 min at room temperature protected from light. Finally, 100 μL stop buffer was added and the plate was incubated for 10 min before reading. The mean fluorescence (RLU) was measured by Varioskan™ LUX multimode microplate reader. The excitation wavelength was 390 nm and the emission was 460 nm.

Regarding β-Galactosidase quantification in electroporated cells, 25,000 electroporated cells with Dbgal were washed with cold PBS twice, and the cell suspension was diluted in 50 μL 1X reaction buffer and analyzed by FACS Blue LacZ beta Galactosidase detection kit following the protocol described above. A control sample consisted of electroporated cells without Dbgal was also measured using the same assay. Finally, the intracellularly delivered β-Galactosidase was calculated by subtracting negative control from the Dbgal electroporated sample.

### HEK293 transfection

HEK293 cells were transiently transfected to express luciferase using Nucleofection. The plasmid used was pLV[Exp]-EGFP/Neo-EF1A>Luciferase (ID: VB900088-2587fmv, VectorBuilder, USA). The plasmid vector was cloned in Stbl3 *E.coli*, and extracted by NucleoBond Xtra Midi kit (Macherey-Nage, Germany). In each transfection sample, 4.3 μg endotoxin-free plasmids were mixed with $2 \times 10^6$ cells suspended in 100 μL electroporation buffer. The mixture was transferred to a 4D-Nucleocuvette™ Vessel, and electroporated with 4D-Nucleofector® X Unit using the pre-optimized program (Code: CM-130). After electroporation, the Nucleocuvette™ Vessel was kept in a cell incubator (+37 °C, 5% CO$_2$ and 95% relative humidity) for 10 min, and then all the cell suspension was transferred to cell culture flasks for incubation. The eGFP expression was checked at 24, and 48 h post transfection by BD LSR-II Cell Analyzer flow cytometer. The luciferase expression was checked at 48 h post transfection by incubating transfected cells with 50 μM D-amLu, and the luminescence was recorded by Varioskan™ LUX multimode microplate reader.

### Cell viability evaluation

The cell viability was evaluated using CellTiter-Glo® luminescent cell viability assay following the manufacturer's protocol. The cell suspension with or without electroporation was added on a 96-well plate at density of $2 \times 10^4$ cells per well in 100 μL cell culture medium. The cell viability was detected at pre-determined time points from the luminescent intensity, in which represent the amount of ATP produced by the viable cells. The assay was carried out according to manufacturer's protocol and the data was recorded by Varioskan™ LUX multimode microplate reader.

### Cellular glutathione concentration evaluation

The cellular glutathione concentration was evaluated using GSH/GSSG-Glo™ assay following the manufacturer's protocol. $1 \times 10^4$ cells with or without electroporation were suspended in 25 μL HBSS buffer and added to wells of a 96-wellplate. Then 25 μL total glutathione lysis reagent or oxidized glutathione lysis reagent was added to the wells and the plate was gently shaked for 5 mins on a plate shaker. Afterwards, Luciferin Generation Reagent was added to the plate (50 μL per well) and the plate was incubated for 30 mins at room temperature. Finally, Luciferin Detection Reagent was added to the plate (100 μL per well), and the plate was shaked for 15 min before reading the luminescence by Varioskan™ LUX multimode microplate reader. Glutathione standard curve was prepared using serial dilution of GSH (0-20 μM) in 25 μL HBSS buffer, followed by the addition of 25 μL total glutathione lysis reagent. Then the same procedure applied as described above for cell suspensions.

### Statistical analysis

Experiments were performed at least in triplicates and all the values were presented as mean ± standard deviation (SD). Statistical significances among different experimental groups were analyzed by using Student's *t*-test (unpaired and two-tailed) and one-way ANOVA analysis. All statistical analysis were carried out using Origin 2022b software.

### Reporting summary

Further information on research design is available in the Nature Portfolio Reporting Summary linked to this article.

## Data availability

All data generated or analyzed during this study are available in the Zenodo repository, with the Digital Object Identifier: 10.5281/zenodo.10927120.

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

## Acknowledgements

S.W. acknowledges the financial support from the Academy of Finland (Grant No. 331106 and 354421), and the European Union (ERC, BioLure, 101115752). M.S. acknowledges the financial support from the Academy of Finland (Grant No. 318422 and 346122). H.A.S. acknowledges financial support from the Academy of Finland (Grant No. 331151) and the UMCG Research Funds. The authors would also like to acknowledge the following core facilities funded by Biocenter Finland: the Light Microscopy Unit of the Institute of Biotechnology for the confocal microscope and the Flow Cytometry Unit for the flow cytometry analyzer. S.W. thanks Prof. Mikko Frilander from the University of Helsinki for the access to Lonza 4D-Nucleofector™, as well as Dr. Esin B Sozer from Northeastern University, Dr. Christos Tapeinos from the University of Manchester, and Dr. Chongyu Zhu from Donghua University for discussions and feedback. Views and opinions expressed are however those of the authors only and do not necessarily reflect those of the European Union or the European Research Council Executive Agency. Neither the European Union nor the granting authority can be held responsible for them.

## Author contributions

S.W. developed the idea of this project. S.W. designed and conducted most of the experiments. M.V.S. contributed to the electroporation experimental design, conducted the native-page gel analysis, and the RNA gel electro-phoresis. S.-P.H. performed the size exclusion chromatography analysis of Dcys-labelled dextran polymers. G.S. performed the elemental analysis of Dcys-labelled dextran polymers. M.S. and H.A.S. provided key resources (cell lines and instrumentation) for this project. S.W. prepared the first draft of the manuscript, which was edited and revised by all authors.

## Competing interests

The authors declare no competing interests.
