## [Peer Review File · Communications Chemistry]

Reviewers' comments:

Reviewer #1 (Remarks to the Author):

The manuscript by S. Wang and co-workers proposes a method for the quantification of intracellular delivery of (bio)molecules of interest using a bioorthogonal luminescent reaction. The approach consists in the labelling of the molecule of interest with D-cysteine, an unnatural amino acid that can be released in the cytoplasm to react with its bioorthogonal partner (NCBT, 6-amino-2-cyanobenzothiazole) and yield D-aminoluciferin. This molecule can be oxidized by luciferase and give rise to a luminescent signal that can be easily monitored and quantified using a plate reader.

The authors provide convincing proofs of the applicability and robustness of this so-called BioLure assay. They show that the reaction cascade can occur in live cells, and that the approach can be used to quantify the intracellular delivery of dextran (as model molecule), lysozyme and RNase A (as relevant biomolecules) after applying cell electroporation.

While the reaction between D-cysteine and NCBT is not new and has been reported before (for example, for real-time monitoring of peptide uptake in live cells and mice, or for real-time imaging of protease activity directly in living animals), the study is of high quality and has potential impact on the field of intracellular delivery of relevant molecules. I commend the authors on the robustness of their experimental approach, which includes validation of the quantification of proteins of interest using traditional quantification methods, and verification of the biological activity of the MOIs after D-cys labelling. In general, the description of the materials and methods is detailed, which indicates that in principle, the work should be easily reproduced by other researchers.

Based on these considerations, I recommend the publication of the work in *Communications Chemistry* after the following revisions:

- 1) The authors should highlight better in the Introduction the added value of the work with respect to the already reported studies using this bioorthogonal reaction between D-cysteine and NCBT.
- 2) Can the authors comment on the applicability of this method to other biomolecules (for example, nucleic acids) or nanoparticles? Also, I have noticed that the three systems chosen as MOIs (Dextran, lysozyme and RNase) have rather similar molecular weights (10, 14 and 13.7 kDa, respectively), yielding similar intracellular concentrations (around 2.5 million of molecules per cell). Was this deliberate? Would the BioLure approach work equally well with smaller or larger molecules?

3) In figure 3, the density of cells in panels 2 (Dex-Dcys) and 3 (Dex-Lcys) seems to be much higher than in panels 1 (control) and 4 (Dex-NC). Why is that?

4) When discussing the effect of DRNase on A375 melanoma cells, the authors mention that “we delivered different amounts of DRNase into cells, by varying the amount of DRNase in the electroporation.”. What are these different amounts?

5) The half-inhibition concentration for the DRNase is of 2.8 ± 1.2 million of DRNase molecules per cell. Even though in this case the authors mention that the cytoplasmatic therapeutic concentration would be in the order of at least one million molecules per cell, I am wondering why the error of the quantification is so large?

Reviewer #2 (Remarks to the Author):

The authors contribute herein a facile protocol using the known luciferin-based bioorthogonal reaction to quantitatively analyze the intracellular delivery of molecule-of-interest via electroporation. The subject is meaningful and interesting. And the proposed method looks effective although similar luciferin-based bioorthogonal systems were reported sometimes. While the presented data are solid enough, the following issues should be addressed before it can be considered for publication in the Communications Chemistry.

1. How much does GSH concentration affect the effectiveness of this system? In vitro and cellular dynamic regulation of GSH should be involved in.

2. How effective is the method on delivery of small molecules? To demonstrate the versatility of system, it should be examined in the delivery of small molecules, such as molecular drugs or fluorophores.

3. As for the results presented in Figure 4, more tests, such as flow cytometry, should be involved to reveal the effect of electroporation on large number of cells.

4. The ladder lane in Figure 4b and 5b should not be cut from other lanes.

Reviewer #3 (Remarks to the Author):

The manuscript titled "Quantitative Analysis of Electroporation-Mediated Intracellular Delivery via Bioorthogonal Luminescent Reaction" by Wang et al. describes a luminescent system, termed

BioLure, based on a bioorthogonal condensation reaction to quantify intracellular delivery. BioLure can estimate the amount of intracellularly delivered molecules following electroporation.

In the BioLure assay, the authors initially labeled target proteins with D-cysteine via a disulfide bond, which can be reduced in the cytoplasm by GSH, thus releasing D-cysteine. NCBT was then introduced to react with D-cysteine, producing D-luciferin and enabling bioluminescent readout detection. The sensitivity and selectivity of the BioLure assay were first tested in both solutions and luciferase-expressing cells, and the reaction conditions were optimized accordingly. Subsequently, the intracellular delivery of model molecules of interest (MOIs), including dextran and lysozyme, was tested and quantified after electroporation using the optimized BioLure assay. Additionally, the authors investigated the precise intracellular delivery dosage necessary for achieving therapeutic outcomes with a functional protein, RNase A.

While the method presented in this work is intriguing and potentially useful, particularly for researchers interested in protein delivery studies, there is a significant concern regarding its novelty. A very similar method was reported five years ago (see ref 35, ACS Chem. Biol. 2019, 14, 2197–2205), wherein the same strategy was employed to evaluate the delivery of peptides into cells. Moreover, that method was even tested in living mice to evaluate its efficiency, thereby diminishing the novelty of the current work.

However, considering that more modern protein delivery methods, apart from conventional electroporation, have now been developed, there is a growing need for the evaluation and comparison of these modern methods. Therefore, in my opinion, extending the evaluation method from peptides to proteins is still worth pursuing if the BioLure assay is utilized to evaluate these newly developed protein delivery methods. Nevertheless, the current form of the manuscript may not be suitable for publication in Commun Chem.

The authors are very grateful to the reviewers for the careful evaluation of the manuscript. A point-by-point response to all the reviews' comments is presented in italics alongside the comments below.

Reviewer #1 (Remarks to the Author):

The manuscript by S. Wang and co-workers proposes a method for the quantification of intracellular delivery of (bio)molecules of interest using a bioorthogonal luminescent reaction. The approach consists in the labelling of the molecule of interest with D-cysteine, an unnatural amino acid that can be released in the cytoplasm to react with its bioorthogonal partner (NCBT, 6-amino-2-cyanobenzothiazole) and yield D-aminoluciferin. This molecule can be oxidized by luciferase and give rise to a luminescent signal that can be easily monitored and quantified using a plate reader. The authors provide convincing proofs of the applicability and robustness of this so-called BioLure assay. They show that the reaction cascade can occur in live cells, and that the approach can be used to quantify the intracellular delivery of dextran (as model molecule), lysozyme and RNase A (as relevant biomolecules) after applying cell electroporation.

While the reaction between D-cysteine and NCBT is not new and has been reported before (for example, for real-time monitoring of peptide uptake in live cells and mice, or for real-time imaging of protease activity directly in living animals), the study is of high quality and has potential impact on the field of intracellular delivery of relevant molecules. I commend the authors on the robustness of their experimental approach, which includes validation of the quantification of proteins of interest using traditional quantification methods, and verification of the biological activity of the MOIs after D-cys labelling. In general, the description of the materials and methods is detailed, which indicates that in principle, the work should be easily reproduced by other researchers.

Based on these considerations, I recommend the publication of the work in Communications Chemistry after the following revisions:

1) The authors should highlight better in the Introduction the added value of the work with respect to the already reported studies using this bioorthogonal reaction between D-cysteine and NCBT.

***Response:** Thank you for the suggestion. We highlighted the value of this work by emphasizing the absolute concentration quantification capability and the use of relatively large protein MOIs in the delivery. The following paragraph has been added in the introduction (Page 6-7) to justify the novelty of the work compared with literature reports of the same reaction.*

"Although similar reactions have been used by others for intracellular peptide internalization studies,¹ only the relative delivery efficiencies were revealed, rather than the absolute concentration of the delivered cargo. The relative intracellular delivery efficiency evaluation methods could provide a comparative perspective of different delivery vehicles, whereas absolute quantification offers essential insights into the therapeutic dose of MOIs. Considering each MOI has its own optimal concentration range to exert its functions without cytotoxicity, the actual amount of intracellularly delivered MOI is

highly valuable for therapeutic applications. In addition, previous reports focused on relatively small MOIs (peptides and small molecules) in cellular uptake studies.¹⁻³ However, this tool has never been explored on large MOIs with complex structures (such as proteins), which offer distinct advantages in therapeutic applications but are difficult to deliver.”

2) Can the authors comment on the applicability of this method to other biomolecules (for example, nucleic acids) or nanoparticles? Also, I have noticed that the three systems chosen as MOIs (Dextran, lysozyme and RNase) have rather similar molecular weights (10, 14 and 13.7 kDa, respectively), yielding similar intracellular concentrations (around 2.5 million of molecules per cell). Was this deliberate? Would the BioLure approach work equally well with smaller or larger molecules?

Response: Thank you for the suggestion. To extend the applicability of this method, we investigated a larger protein MOI, β -Galactosidase (465kDa, tetramer). We chose this protein because it is an enzyme and there are complementary enzymatic activity kits available to verify the quantification after electroporation-mediated intracellular delivery.

So first we conjugated Dcys on β -Galactosidase (bGal), and then characterized the Dcys-labelled bGal (DbGal). Due to the complex structure, we did not manage to separate the protein on Native-Page gel despite trying different gel compositions and running voltages. But in the reducing SDS-PAGE gel, we managed to get clear bands of both bGal and DbGal (Figure S11). The original bGal showed multiple bands, but the main band was 125 kDa, corresponding to one subunit of the protein. Similarly, DbGal also showed multiple bands with the main band at the same molecular weight. This means after reduction, DbGal could be restored to its original form.

Then we tested DbGal in the electroporation-mediated intracellular delivery. As shown in Figure S12, the real-time luminescence signal output followed a similar trend as Dcys-labelled dextran and lysozymes. Compared with lysozymes and dextran, the final estimated delivered DbGal was lower, c.a., 0.28 ± 0.04 million per cell (Figure 4d). The lower delivery efficiency could be attributed to the large size of bGal, but other factors such as charge, structure, membrane association may also affect the electroporation-mediated cell entry. The intracellular delivery of DbGal was also verified by FACS Blue LacZ beta Galactosidase Detection Kit, which allows for quantifying the protein amount in intact cells after electroporation. We obtained similar delivery results from enzymatic activity assays (0.32 ± 0.04 million per cell), compared with BioLure results in Figure 4d. Thus, it is concluded that BioLure assay could be applied to quantify the delivery of protein MOIs of various sizes.

Relevant results (Figure 4d, Figure S11, Figure S12) and discussions have been added in the revised manuscript (Page 17-18) and the revised supporting information.

3) In figure 3, the density of cells in panels 2 (Dex-Dcys) and 3 (Dex-Lcys) seems to be much higher than in panels 1 (control) and 4 (Dex-NC). Why is that?

Response: *We appreciate the careful observation. Such density difference is due to the cell sedimentation variations from sample to sample. All the confocal images were taken within 30 mins after electroporation, when the cells were still suspended in the buffer. Therefore, the sedimentation affected the cell density observed in the scope, and if some cells were not fully sedimented and out of the focal plane, they were absent from the image.*

4) When discussing the effect of DRNase on A375 melanoma cells, the authors mention that “we delivered different amounts of DRNase into cells, by varying the amount of DRNase in the electroporation.”. What are these different amounts?

Response: *The amount of DRNase in the electroporation ranges from 0 to 20 μg per sample. We added this information in the revised manuscript (Page 20).*

5) The half-inhibition concentration for the DRNase is of 2.8 ± 1.2 million of DRNase molecules per cell. Even though in this case the authors mention that the cytoplasmatic therapeutic concentration would be in the order of at least one million molecules per cell, I am wondering why the error of the quantification is so large?

Response: *We admit that the half-inhibition concentration estimation was not optimal, and the large error of the half-inhibition concentration ($K_{1/2}$) was mainly due to the fitting, as it was calculated from the fitting curve shown in Figure 5d. In the fitting, there are only 7 data points, so the fitting algorithm may not have enough information to accurately estimate the parameters of the Michaelis-Menten equation. This is also reflected by the non-ideal R-square value (0.9). Consequently, there is high variability in the estimated parameters i.e., the calculated half-inhibition value. Furthermore, all the original data points have various standard deviations. Especially, the intracellularly delivered DRNase (x-axis) varied from sample to sample, even with the same amount of DRNase in the electroporation. The cell growth inhibition percentage (y-axis) also had obvious variability when the inhibition percentage was low. This suggests the original data consistency may contribute to the fitting errors. Such limitation could be overcome by increasing the number of data points and refining the fitting methods in the future work.*

We added the relevant discussions about the large standard deviation of $K_{1/2}$ and the limitation of the study in the revised manuscript (Page 21).

Reviewer #2 (Remarks to the Author):

The authors contribute herein a facile protocol using the known luciferin-based bioorthogonal reaction to quantitatively analyze the intracellular delivery of molecule-of-interest via electroporation. The subject is meaningful and interesting. And the proposed method looks effective although similar luciferin-based bioorthogonal systems were reported sometimes. While the presented data are solid enough, the following issues should be addressed before it can be considered for publication in the Communications Chemistry.

1. How much does GSH concentration affect the effectiveness of this system? In vitro and cellular dynamic regulation of GSH should be involved in.

Response: We agree with the reviewer that the GSH concentration is an important factor that could potentially affect the effectiveness of the system. According to literature reports, intracellular GSH concentration is within the millimolar range^{4,5}, which is far more excessive than the D-cysteine concentration used in this system (in the micromolar range). Thus, we assume that the GSH concentration is not the limiting factor of the reaction.

To verify the intracellular GSH concentration of A375 cells used in this study, we measured it using a commercial kit (GSH/GSSG-Glo™ assay, Promega). We evaluated both healthy cells and cells right after electroporation. The electroporated cells represent an extreme scenario when cells are exposed to transient membrane damages and subjected to GSH depletion.⁶ The results suggested the GSH concentration within healthy A375 cells is 4.97 mM, which is similar to literature reports of cancer cell intracellular GSH concentrations measured by various other methods (e.g., 1.4 mM in HepG2 cells measured by HPLC⁷; 3.9-5.4 mM in HeLa cells measured by fluorescent probes^{8,9}; 6.1 mM in HeLa cells measured by a single-cell nanopore sensor¹⁰). After electroporation, the GSH concentration decreased to 1.37 mM. It is still in micromolar range and much higher than the D-cysteine concentration. Based on these results, we believe the intracellular GSH concentration is significantly higher than D-cysteine to be analyzed by the BioLure assay.

There is a technical limitation in the currently used GSH concentration evaluation assay, which requires cell lysis. Therefore, we could not monitor real-time GSH concentrations in live cells by this assay. However, we would like to further investigate this issue in the future work. Relevant discussions have been added in the revised manuscript (Page 8-9).

2. How effective is the method on delivery of small molecules? To demonstrate the versatility of system, it should be examined in the delivery of small molecules, such as molecular drugs or fluorophores.

Response: Thank you for the comment. In this manuscript, our method is developed for characterizing the intracellular delivery of membrane-impermeable biomacromolecules instead of small molecules, because biotherapeutics show great potential to treat undruggable diseases and it is difficult to know the

actual intracellular delivery dosage required for satisfying therapeutic outcomes. Therefore, we simply focused on different kinds of biomacromolecules (dextran and proteins), and the small molecule delivery is outside the scope of the research.

However, to further demonstrate the versatility of the system, we included another protein payload with higher molecular weight (beta-galactosidase, bGal) in the revised manuscript. As shown in Figure 4d and Figure S12 in the revised supporting information, D-cysteine labelled bGal (DbGal) could be intracellularly delivered by electroporation, and the amount of DbGal delivered could be quantified by the BioLure assay. The results obtained showed consistency with the complementary method (FACS Blue LacZ beta Galactosidase detection kit). We believe the inclusion of DbGal data demonstrate that the BioLure assay applies to versatile protein payloads with a broad range of molecular weights.

Relevant results (Figure 4d, Figure S11, Figure S12) and discussions have been added in the revised manuscript (Page 17-18) and the revised supporting information.

3. As for the results presented in Figure 4, more tests, such as flow cytometry, should be involved to reveal the effect of electroporation on large number of cells.

Response: Thank you for the suggestion. We have added the flow cytometry results of Alexa Fluor 647 labelled Dcys-Dex electroporation in Figure S8 (supporting information). As shown in the histogram, there is a wide distribution of fluorescence intensity within the electroporated cell populations. This is due to the heterogeneous nature of the bulk electroporation technique.¹¹ When cells are exposed to the electro current field, the transient pore formation varies from cell to cell, probably due to the differences in the local electronic field on individual cell level. This eventually leads to different intracellular delivery efficiencies. Relevant discussions have been added in the revised manuscript (Page 14).

4. The ladder lane in Figure 4b and 5b should not be cut from other lanes.

Response: Thank you for pointing out this issue. We removed the cut gel figures from Figure 4 and Figure 5. Instead, we put the whole gel image in the supplementary information (Figure S11).

Reviewer #3 (Remarks to the Author):

The manuscript titled "Quantitative Analysis of Electroporation-Mediated Intracellular Delivery via Bioorthogonal Luminescent Reaction" by Wang et al. describes a luminescent system, termed BioLure, based on a bioorthogonal condensation reaction to quantify intracellular delivery. BioLure can estimate the amount of intracellularly delivered molecules following electroporation. In the BioLure assay, the authors initially labeled target proteins with D-cysteine via a disulfide bond, which can be reduced in the cytoplasm by GSH, thus releasing D-cysteine. NCBT was then introduced to

react with D-cysteine, producing D-luciferin and enabling bioluminescent readout detection. The sensitivity and selectivity of the BioLure assay were first tested in both solutions and luciferase-expressing cells, and the reaction conditions were optimized accordingly. Subsequently, the intracellular delivery of model molecules of interest (MOIs), including dextran and lysozyme, was tested and quantified after electroporation using the optimized BioLure assay. Additionally, the authors investigated the precise intracellular delivery dosage necessary for achieving therapeutic outcomes with a functional protein, RNase A.

While the method presented in this work is intriguing and potentially useful, particularly for researchers interested in protein delivery studies, there is a significant concern regarding its novelty. A very similar method was reported five years ago (see ref 35, ACS Chem. Biol. 2019, 14, 2197–2205), wherein the same strategy was employed to evaluate the delivery of peptides into cells. Moreover, that method was even tested in living mice to evaluate its efficiency, thereby diminishing the novelty of the current work.

However, considering that more modern protein delivery methods, apart from conventional electroporation, have now been developed, there is a growing need for the evaluation and comparison of these modern methods. Therefore, in my opinion, extending the evaluation method from peptides to proteins is still worth pursuing if the BioLure assay is utilized to evaluate these newly developed protein delivery methods. Nevertheless, the current form of the manuscript may not be suitable for publication in Commun Chem.

Response: We apologize that the novelty was not clarified in the original version of the manuscript. In the revised introduction, we elaborated the novelty of our work compared with previous publications using the same reaction:

“Although similar reactions have been used by others for intracellular peptide internalization studies,¹ only the relative delivery efficiencies were revealed, rather than the absolute concentration of the delivered cargo. The relative intracellular delivery efficiency evaluation methods could provide a comparative perspective of different delivery vehicles, whereas absolute quantification offers essential insights into the therapeutic dose of MOIs. Considering each MOI has its own optimal concentration range to exert its functions without cytotoxicity, the actual amount of intracellularly delivered MOI is highly valuable for therapeutic applications. In addition, previous reports focused on relatively small MOIs (peptides and small molecules) in cellular uptake studies.^{1–3} However, this tool has never been explored on large MOIs with complex structures (such as proteins), which offer distinct advantages in therapeutic applications but are difficult to deliver.”

Overall, our method demonstrated the absolute concentration quantification capability of relatively large protein MOIs. The paragraph above has been added in the introduction (Page 6-7).

Regarding extending the delivery methods, we really appreciate this constructive comment. It is indeed worth investigating the applicability of the BioLure assay using different delivery methods, e.g., cell penetrating peptides- or nanoparticles-mediated intracellular protein delivery.¹² Such results would be

beneficial to reveal the protein therapeutics dosage in a more clinical-relevant setting. However, the aim of the present study was to provide proof-of-concept data for the developed BioLure assay and to demonstrate that there is potential for the future studies. We believe that the results presented in this manuscript using the conventional but robust electroporation method are convincing enough to realize the aim. Based on these results, we are currently investigating the possibility to use more sophisticated delivery methods. However, when we change the delivery methods, the major parameters of the BioLure assay (i.e., NCBT concentration, cell density, reaction buffer, etc.) need further optimizations and it requires significantly amount of work. Then from our opinion, the application of BioLure assay in new delivery methods would be by itself worth of a separated publication.

References:

1. Karatas, H. *et al.* Real-Time Imaging and Quantification of Peptide Uptake in Vitro and in Vivo. *ACS Chem. Biol.* **14**, 2197–2205 (2019).
2. Bazhin, A. A. *et al.* A Universal Assay for Aminopeptidase Activity and Its Application for Dipeptidyl Peptidase-4 Drug Discovery. *Anal. Chem.* **91**, 1098–1104 (2019).
3. Godinat, A., Bazhin, A. A. & Goun, E. A. Bioorthogonal chemistry in bioluminescence imaging. *Drug Discov. Today* **23**, 1584–1590 (2018).
4. Camera, E. & Picardo, M. Analytical methods to investigate glutathione and related compounds in biological and pathological processes. *Journal of Chromatography B: Analytical Technologies in the Biomedical and Life Sciences* vol. 781 at [https://doi.org/10.1016/S1570-0232\(02\)00618-9](https://doi.org/10.1016/S1570-0232(02)00618-9) (2002).
5. Giustarini, D. *et al.* Assessment of glutathione/glutathione disulphide ratio and S-glutathionylated proteins in human blood, solid tissues, and cultured cells. *Free Radical Biology and Medicine* vol. 112 at <https://doi.org/10.1016/j.freeradbiomed.2017.08.008> (2017).
6. Kulbacka, J. *et al.* Nanoelectropulse delivery for cell membrane perturbation and oxidation in human colon adenocarcinoma cells with drug resistance. *Bioelectrochemistry* **150**, (2023).
7. Dominick, P. K., Cassidy, P. B. & Roberts, J. C. A new and versatile method for determination of thiolamines of biological importance. *J. Chromatogr. B Biomed. Sci. Appl.* **761**, (2001).
8. Liu, Z. *et al.* A Reversible Fluorescent Probe for Real-Time Quantitative Monitoring of Cellular Glutathione. *Angew. Chemie - Int. Ed.* **56**, (2017).
9. Liu, H. *et al.* A ratiometric fluorescent probe for real-time monitoring of intracellular glutathione fluctuations in response to cisplatin. *Chem. Sci.* **11**, (2020).
10. Hu, P., Zhang, Y., Wang, D., Qi, G. & Jin, Y. Glutathione Content Detection of Single Cells under Ingested Doxorubicin by Functionalized Glass Nanopores. *Anal. Chem.* **93**, (2021).
11. Stewart, M. P., Langer, R. & Jensen, K. F. Intracellular Delivery by Membrane Disruption: Mechanisms, Strategies, and Concepts. *Chem. Rev.* **118**, 7409–7531 (2018).
12. Zhang, Y., Røise, J. J., Lee, K., Li, J. & Murthy, N. Recent developments in intracellular protein delivery. *Curr. Opin. Biotechnol.* **52**, 25–31 (2018).

REVIEWERS' COMMENTS:

Reviewer #1 (Remarks to the Author):

The authors have fully addressed all my comments and performed additional experiments to further demonstrate the versatility of the BioLure assay. I believe the manuscript is now ready for publication in Communications Chemistry without further changes.

Reviewer #2 (Remarks to the Author):

Thank you for the effort. The authors have addressed my concerns. The manuscript can be accepted as it is.

Reviewer #3 (Remarks to the Author):

The authors have added additional description in the introduction section to clarify the novelty of their work compared to previously reported methods. While I appreciate this effort, I am not persuaded by this revision. The authors emphasize that their method provides quantification capability, but such capability could also be achieved by the previously reported method. I do not see any superior points of BioLure technique in this paper.

In my opinion, there should be at least one additional experiment to compare at least two protein delivery methods, demonstrating the evaluation capability of BioLure. Therefore, I cannot recommend the publication of this paper in its current form.